# The subcortical and neurochemical organization of the ventral and dorsal attention networks

Pedro Nascimento Alves[1,2 ✉], Stephanie J. Forkel[3,4,5,6], Maurizio Corbetta [7,8,9,10] & Michel Thiebaut de Schotten [3,11 ✉]

Attention is a core cognitive function that filters and selects behaviourally relevant information in the environment. The cortical mapping of attentional systems identified two segregated networks that mediate stimulus-driven and goal-driven processes, the Ventral and the Dorsal Attention Networks (VAN, DAN). Deep brain electrophysiological recordings, behavioral data from phylogenetic distant species, and observations from human brain pathologies challenge purely corticocentric models. Here, we used advanced methods of functional alignment applied to resting-state functional connectivity analyses to map the subcortical architecture of the Ventral and Dorsal Attention Networks. Our investigations revealed the involvement of the pulvinar, the superior colliculi, the head of caudate nuclei, and a cluster of brainstem nuclei relevant to both networks. These nuclei are densely connected structural network hubs, as revealed by diffusion-weighted imaging tractography. Their projections establish interrelations with the acetylcholine nicotinic receptor as well as dopamine and serotonin transporters, as demonstrated in a spatial correlation analysis with a normative atlas of neurotransmitter systems. This convergence of functional, structural, and neurochemical evidence provides a comprehensive framework to understand the neural basis of attention across different species and brain diseases.

[1] Laboratório de Estudos de Linguagem, Centro de Estudos Egas Moniz, Faculdade de Medicina, Universidade de Lisboa, Lisboa, Portugal. [2] Serviço de Neurologia, Departamento de Neurociências e Saúde Mental, Hospital de Santa Maria, CHULN, Lisboa, Portugal. [3] Brain Connectivity and Behaviour Laboratory, Sorbonne University, Paris, France. [4] Donders Institute for Brain Cognition Behaviour, Radboud University, Thomas van Aquinostraat 4, 6525GD Nijmegen, the Netherlands. [5] Centre for Neuroimaging Sciences, Department of Neuroimaging, Institute of Psychiatry, Psychology and Neuroscience, King's College London, London, UK. [6] Departments of Neurosurgery, Technical University of Munich School of Medicine, Munich, Germany. [7] Clinica Neurologica, Department of Neuroscience, University of Padova, Padova, Italy. [8] Padova Neuroscience Center (PNC), University of Padova, Padova, Italy. [9] Venetian Institute of Molecular Medicine, VIMM, Padova, Italy. [10] Department of Neurology, Radiology, Neuroscience Washington University School of Medicine, St.Louis, MO, USA. [11] Groupe d'Imagerie Neurofonctionnelle, Institut des Maladies Neurodégénératives-UMR 5293, CNRS, CEA, University of Bordeaux, Bordeaux, France. ✉email: pedronascimentoalves@gmail.com; michel.thiebaut@gmail.com

"Everyone knows what attention is. It is the taking possession by the mind, in clear and vivid form, of one out of what seem several simultaneously possible objects or trains of thought."[1].

Everything we see, feel, or smell is an illusion elaborated by our brain circuits. However, the brain's capacity is limited. This requires mechanisms for the selection of the most relevant information. The ensemble of cognitive and neural processes involved in capacity limitation and selection underlies 'attention' as defined by James[1]. Behavioral studies have distinguished orienting of attention into a slow, strategic, goal-directed, and voluntary component versus a swift, unexpected, bottom-up, and automatic component[2,3]. Task-related functional neuroimaging (fMRI) studies segregated these two attentional processes anatomically into a dorsal and ventral attentional network[4]. The dorsal attention network (DAN) encodes and maintains preparatory signals and modulates top-down sensory (visual, auditory, olfactory and somatosensory) regions.

In contrast, the ventral attention network (VAN) is recruited when attention is re-oriented to novel behaviorally relevant events. Classical core regions of the DAN are the intraparietal sulcus, the superior parietal lobe, and the frontal eye fields. The DAN is considered to have no hemispheric lateralization[5–9]. In contrast, the temporoparietal junction and the ventrolateral prefrontal cortex constitute the central regions of the VAN. Evidence demonstrates that the VAN is right-lateralized[7,8,10]. Within their respective networks, DAN and VAN regions have synchronous fMRI signal oscillations at rest[8,11–18]. Thanks to this synchronization, the two networks have consistently been identified and segregated in resting-state fMRI cortical parcellations[14,16–18], although their taxonomy has not always been homogenous in the literature[19,20]. Hence, the DAN and VAN are organized as independent networks even in the absence of task signals. However, their synchronization can change according to task demands, and they can be acting jointly or separately[21,22]. Furthermore, DAN and VAN task activations and synchronization levels are modified by focal lesions and correlate with behavioral deficits[23–30].

Yet, electrical recording, pathological observations, and phylogenetic comparisons demonstrate that the neuroanatomical framework of attentional mechanisms should extend well beyond a corticocentric model. Electrical recordings in primates showed that subcortical structures have a crucial role in the neural mechanisms of attention. For instance, inactivation of the superior colliculus during motion-change detection markedly disturbs visual attention without affecting the neuronal activity in the visual cortex[31]. Attentional states also modulate the thalamic pulvinar nuclei[32,33] and neuronal discharge patterns in the locus coeruleus[34,35]. Pathophysiological data from human brain disease supports the critical relevance of deep brain nuclei. Neglect is a clinical syndrome characterized by pathological hemispatial inattention[36] and can arise from subcortical lesions in the pulvinar, striatum, or superior colliculus[37–40]. Patients with attention deficit hyperactivity disorder also present alterations beyond the cortex[28], such as in the pulvinar, which is influenced by the severity of the disease and the use of stimulants[41].

Additionally, distant phylogenetic species, such as pigeons, have markedly different cortical morphologies but exhibit attention errors and reaction times similar to humans[42]. With close mammals, such as macaques, relevant functional attention dissimilarities have been described at the cortical level, including the complete absence of a VAN[43]. Hence, a core phylogenetically relevant subcortical network of areas appears to support the orientation of attention that has been mostly disregarded in the functional neuroimaging literature because of

limited field strength or issues arising from average group alignments. Average group alignments of functional neuroimaging maps exclusively based on structural landmarks might typically fail to represent an accurate functional network due to interindividual differences[44,45]. Specifically, subcortical nuclei are prone to structural misalignment due to their small size, poor contrast in structural MRI, and intersubject cytoarchitectonic variability[46–49]. In contrast, advanced methods of functional alignment improve structural-functional correspondence across participants[50–53]. Further, surface interindividual alignment based on morphological features, such as cortical folding, fairly aligns unimodal cortical areas, such as the primary visual and motor cortices, but poorly overlaps higher-order cortical areas[50,54]. Methods of functional alignment based on fMRI signals during cognitive activation paradigms[55,56] and resting-state fMRI connectivity patterns[52,57] provided better function matching and have also been used for cross-species functional comparisons[58]. Functional alignment is different from hyper-alignment techniques that project shared neural information beyond the three-dimensional anatomical space, i.e., in high-dimensional spaces[59–61]. At the subcortical level, our team also demonstrated that functional alignment methods can optimize the group-level mapping of functional networks, improving functional correlations and uncovering a network's deep brain nuclei components[62]. However, this method has never been applied to explore the subcortical anatomy of the VAN and the DAN.

Delineating the subcortical components of the DAN and the VAN would allow us to revisit their underlying circuitry through diffusion-weighted imaging tractography that enables in vivo reconstruction of associative, commissural, and projection white-matter tracts[63–65]. A clearer characterization of the DAN and VAN circuitry will help to better understand brain interactions in healthy and pathological brains[66,67].

Subcortical structures also play a critical role within the neurotransmitter systems. Brainstem nuclei are the primary sources of neurotransmitter synthesis and send axonal projections to the cortex and the basal ganglia. The basal ganglia are central targets of the neurotransmitter axonal projections and mediate their physiological effects. Yet the neurochemistry of the DAN and the VAN is limited to primate studies. These studies reported a noradrenergic innervation of regions of the primate attention networks, including the temporoparietal junction and the frontal lobe[68–70]. Noradrenaline has been proposed as a critical trigger for the reorientation of attention[8,70]. However, despite its essential neuroscientific and medical importance[28], the neurochemical signatures of the VAN and the DAN have never been contrasted in humans. Such an endeavor is now possible thanks to the macroscale mapping of the neurotransmitter receptors and transporters in humans by means of positron emission tomography (PET) and single-photon emission computerized tomography (SPECT) scans[71]. Accordingly, a normative atlas of nine neurotransmitter systems aligned in the MNI space is now openly available and allows the investigation of the neurochemical signature of brain circuits[72–76].

Therefore, we explored the subcortical anatomy of attention networks by aligning the individual resting-state functional maps of the VAN and the DAN in a common functional space. Based on previous electrical recordings, pathological observations, and phylogenetic reports, we hypothesized that basal ganglia and brainstem nuclei, namely the pulvinar, the striatum, the superior colliculi, and the locus coeruleus, are core phylogenetically relevant and functional constituents of the attention networks. Finally, an optimized model of the VAN and the DAN was proposed together with their structural, functional, graph centrality, and neurochemical signature.

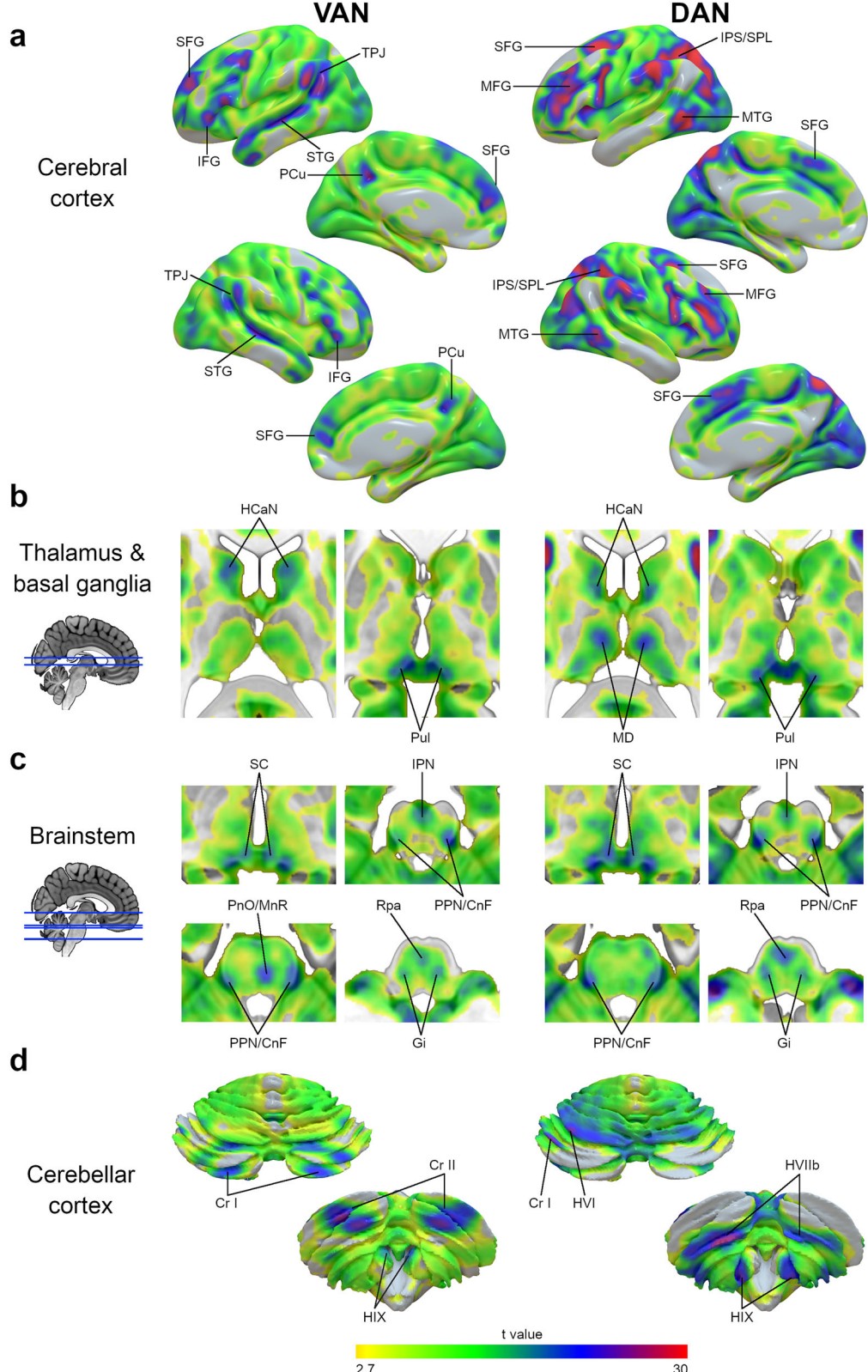

## Results

**VAN anatomical map**. The statistical map of the VAN, after functional alignment, is represented in Fig. 1 (left column).

At the cerebral cortical level, the peaks of statistical association were observed in the temporoparietal junction, the inferior frontal gyrus, the anterior part of the superior frontal gyrus, and the superior temporal gyrus (Fig. 1a). Additionally, peaks of

statistical association were also present in the crus I, crus II and superior IX cerebellar cortex (Fig. 1d).

A high statistical association was present at the thalamus and basal ganglia level in the head of caudate nuclei and the pulvinar (Fig. 1b). In the brainstem, a high statistical association was observed in voxels overlapping with the superior colliculi, the interpeduncular nucleus, and the pedunculopontine-cuneiform

**Fig. 1 VAN and DAN maps after functional alignment.** VAN (left) and DAN (right) maps after functional alignment at different anatomical levels, namely the cerebral cortical surface (**a**), subcortical thalamus and basal ganglia (**b**), brainstem (**c**), and cerebellar cortical surface (**d**). The color gradient represents the t-value distribution ($n = 110$). CnF cuneiform nucleus, Cr I cerebellar crus I lobule, Cr II cerebellar crus II lobule, DAN dorsal attention network, Gi gigantocellular nucleus, HCaN head of caudate nucleus, HIIb cerebellar lobule IIb, HVI cerebellar lobule VI, HIX cerebellar lobule IX (cerebellar tonsils), IFG inferior frontal gyrus, IPN interpeduncular nucleus, IPS intraparietal sulcus, MnR median raphe nucleus, MD mediodorsal nucleus of the thalamus, MTG middle temporal gyrus, PCu precuneus, PnO nucleus pontis oralis, PPN pedunculopontine nucleus, Pul pulvinar, Rpa raphe pallidus nucleus, SC superior colliculus, SFG superior frontal gyrus, SPL superior parietal lobule, STG superior temporal gyrus, TPJ temporoparietal junction, VAN ventral attention network.

**Table 1 MNI coordinates of the VAN subcortical regions' centers of gravity.**

| Regions of interest | MNI (X) | MNI (Y) | MNI (Z) |
|---|---|---|---|
| HCaN L | −11 | 8 | 13 |
| Pul L | −4 | −30 | 1 |
| SC L | −9 | −31 | −3 |
| PPN/CnF L | −14 | −29 | −25 |
| Gi L | −10 | −25 | −36 |
| Cr I L | −31 | −73 | −31 |
| Cr II L | −21 | −79 | −42 |
| IPN | 1 | −19 | −21 |
| MnR | −3 | −29 | −28 |
| Rpa | 1 | −28 | −43 |
| HCaN R | 13 | 11 | 12 |
| Pul R | 6 | −29 | 1 |
| SC R | 13 | −30 | −3 |
| PPN/CnF R | 13 | −31 | −25 |
| Gi R | 11 | −24 | −35 |
| Cr I R | 30 | −74 | −30 |
| Cr II R | 24 | −79 | −41 |

*CnF cuneiform nucleus, Cr I cerebellar crus I lobule, Cr II cerebellar crus II lobule, Gi gigantocellular nucleus, HCaN head of caudate nucleus, IPN interpeduncular nucleus, L left, MnR median raphe nucleus, PnO nucleus pontis oralis, PPN pedunculopontine nucleus, Pul pulvinar, R right, Rpa raphe pallidus nucleus, SC superior colliculus.*

**Table 2 MNI coordinates of the DAN subcortical regions' centers of gravity.**

| Regions of interest | MNI (X) | MNI (Y) | MNI (Z) |
|---|---|---|---|
| HCaN L | −10 | 4 | 10 |
| MD L | −9 | −18 | 8 |
| Pul L | −4 | −30 | 1 |
| SC L | −8 | −31 | −3 |
| PPN/CnF L | −14 | −30 | −25 |
| Gi L | −10 | −24 | −35 |
| Cr I L | −39 | −64 | −29 |
| HVI L | −25 | −62 | −24 |
| HVIIb L | −24 | −66 | −49 |
| HIX L | −11 | −51 | −50 |
| IPN | 1 | −19 | −21 |
| Rpa | 1 | −28 | −42 |
| HCaN R | 12 | 7 | 10 |
| MD R | 9 | −16 | 8 |
| Pul R | 6 | −29 | 1 |
| SC R | 11 | −30 | −3 |
| PPN/CnF R | 14 | −30 | −25 |
| Gi R | 11 | −25 | −34 |
| HVIIb R | 25 | −68 | −49 |
| HIX R | 11 | −53 | −51 |

*CnF cuneiform nucleus, Cr I cerebellar crus I lobule, Gi gigantocellular nucleus, HCaN head of caudate nucleus, HVI cerebellar lobule VI, HIX cerebellar lobule IX (cerebellar tonsils), IPN interpeduncular nucleus, L left, MD mediodorsal nucleus of the thalamus, PPN pedunculopontine nucleus, Pul pulvinar, R right, Rpa raphe pallidus nucleus, SC superior colliculus.*

nuclei complex pontis oralis, the gigantocellular nuclei, the raphe pallidus, and median nuclei (Fig. 1c). Table 1 represents the centers of gravity coordinates of the subcortical regions of interest. The VAN statistical and correlation maps are available at https://neurovault.org/collections/XONZLGPJ/.

**DAN anatomical map.** The statistical map of the DAN, after functional alignment, is represented in Fig. 1 (right column).

The peaks of the statistical association at the cerebral cortical level were in the intraparietal sulcus and superior parietal lobule, in the middle and superior frontal gyrus, and in the posterior part of the middle temporal gyrus (Fig. 1a). Peaks of statistical association were also present in the cerebellar cortex's areas VIIb, inferior IX, left VI, and left I (Fig. 1d).

At the thalamus and basal ganglia level, areas with a high statistical association were located in the head of caudate nuclei and the thalamic pulvinar and mediodorsal nuclei (Fig. 1b). High statistical associations also included voxels overlapping the superior colliculi, the interpeduncular nucleus, the pedunculopontine-cuneiform nuclei complex, the gigantocellular nuclei, and the raphe pallidus nuclei in the brainstem (Fig. 1c). Table 2 represents the centers of gravity of the subcortical regions of interest. The DAN statistical and correlation maps are available at https://neurovault.org/collections/XONZLGPJ/.

The conjunction analysis showed that most of the subcortical peaks of statistical association were shared by both networks (Fig. 2), explicitly overlapping the pulvinar, the superior colliculi,

the interpeduncular nuclei, the pedunculopontine-cuneiform nuclei complex, the gigantocellular nuclei, and the raphe pallidus nuclei.

**Structural and functional connectivity of the VAN nodes.** The structural connectivity map of the VAN is represented in Fig. 3a.

The cortical regions of the VAN were connected by the third branch of the Superior Longitudinal Fasciculus (SLF III) and the uncinate fasciculus (Fig. 3a). Fronto-pulvinar and tecto-pulvinar projections established the connections with or between subcortical structures (Fig. 3a). The node-to-node structural and functional connectivity patterns are represented in Fig. 3b.

The maps of the VAN ROIs and the structural connectivity analysis are available at https://neurovault.org/collections/XONZLGPJ/.

**Structural and functional connectivity of DAN nodes.** The structural connectivity map of the DAN is represented in Fig. 3c.

The cortical regions of the DAN established connections through the first branch of the Superior Longitudinal Fasciculus (SLF I, Fig. 3c). Fronto-pulvinar, parieto-pulvinar, and tecto-pulvinar projections mediated the links with or between subcortical structures (Fig. 3c).

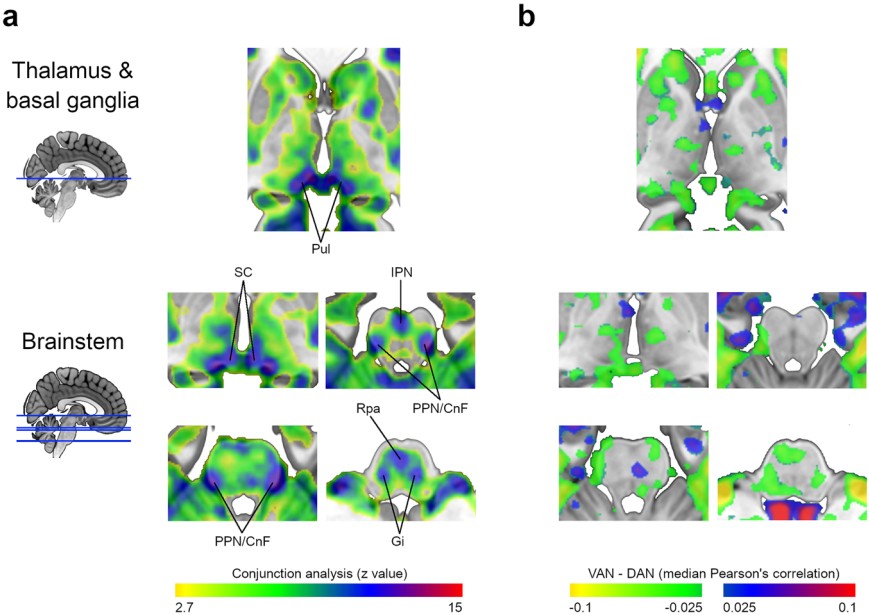

**Fig. 2 VAN and DAN maps similarity. a** Conjunction analysis of the VAN and DAN statistical maps at the thalamus, basal ganglia, and brainstem levels ($n = 110$). **b** Difference map resulting from the subtraction of the median DAN Pearson's correlation map from the VAN. CnF cuneiform nucleus, DAN dorsal attention network, Gi gigantocellular nucleus, IPN interpeduncular nucleus, PPN pedunculopontine nucleus, Pul pulvinar, Rpa raphe pallidus nucleus, SC superior colliculus, VAN ventral attention network. (see maps at https://neurovault.org/collections/XONZLGPJ/).

The node-to-node structural and functional connectivity patterns are shown in Fig. 3d.

The maps of the DAN ROIs and the structural connectivity analysis are available at https://neurovault.org/collections/XONZLGPJ/.

**Lateralization assessment**. Figure 4 illustrates the hemispheric distribution of the structural and functional connectivity measures of the VAN and the DAN.

The structural connectivity connecting the VAN was significantly larger in the right hemisphere than in the left (right hemisphere 18.7[16.5,21.1]cm³, left hemisphere 17.0[15.3,19.7]cm³; $p$ value <0.001). Pearson's correlations were not different between the right and left VANs (right hemisphere 0.185[0.162,0.218], left hemisphere 0.188[0.160,0.222]; $p$ value = 0.125).

The DAN's structural connectivity was also significantly larger in the right hemisphere (right hemisphere 33.6[30.0,36.6]cm³, left hemisphere 30.2[28.0,32.9]cm³; $p$ value <0.001). Pearson's correlations were significantly higher in the left hemisphere than in the right (right hemisphere 0.240(0.049), left hemisphere 0.246(0.050); $p$ value <0.001).

**Graph theory analysis**. Figure 5a illustrates the graph theory representation of the VAN and DAN structural connectivity.

The subcortical structures with the highest median betweenness centrality in the VAN were in the right pulvinar and the left caudate nucleus head (the second and the third highest of all nodes, respectively). The highest median degree of centrality was in the interpeduncular nucleus and the left pedunculopontine-cuneiform nuclei complex (the first and the second highest of all nodes, respectively).

In the DAN, the subcortical structures with the highest median betweenness centrality were the raphe pallidus nucleus and the right mediodorsal nucleus of the thalamus (the first and the seventh highest of all nodes, respectively). The highest median degree of centrality was the raphe pallidus nucleus and the left superior colliculus (the first and the second highest of all nodes, respectively).

Overall, the subcortical structures had high centrality values in both networks. The betweenness centrality and degree centrality values of all nodes in the VAN and the DAN are detailed in Supplementary Tables 3, 4. The anatomical models of the VAN and DAN are illustrated in Fig. 5b.

**Correlation with the neurotransmitter system**. The brainstem nuclei identified in the VAN anatomical map that synthesize neurotransmitters are the pedunculopontine nuclei (cholinergic, glutamatergic, and GABAergic; Benarroch[72]), the cuneiform nuclei (glutamatergic and GABAergic; Chang et al.[73]), the gigantocellular nucleus (glutamatergic and GABAergic; Martin et al.[74]), the raphe nucleus (serotonergic; Van De Kar and Lorens[75]), and the raphe pallidus nucleus (serotonergic; Heym et al.[76]). The brainstem nuclei identified in the DAN anatomical map synthesizing neurotransmitters are the pedunculopontine, the cuneiform, the gigantocellular nuclei, and the raphe pallidus nucleus.

The spatial correlations of these brainstem nuclei structural projections with the neurotransmitter systems are represented in Fig. 6. The distributions of acetylcholine α4β2 nicotinic receptors, dopamine transporters, and serotonin transporters were positively correlated with the distribution of VAN and DAN brainstem projections ($p < 0.001$; Fig. 6a). The scatterplots representing the distributions of the significantly correlated systems are presented in Fig. 6b. Acetylcholine α4β2 nicotinic receptors and serotonin transporters had a higher spatial correlation with the VAN than with the DAN, whereas dopamine transporters had a higher spatial correlation with the DAN ($p < 0.001$).

The supplemental pairwise correlation analyses between the average VAN and DAN structural projection maps and the neurotransmitter maps revealed similar results: the VAN had a significant positive spatial correlation with acetylcholine α4β2 nicotinic receptors and acetylcholine, dopamine, noradrenaline, and serotonin transporters (Supplementary Table 5); the DAN had a significant positive spatial correlation with acetylcholine α4β2 nicotinic receptors and acetylcholine, dopamine and noradrenaline transporters (Supplementary Table 6).

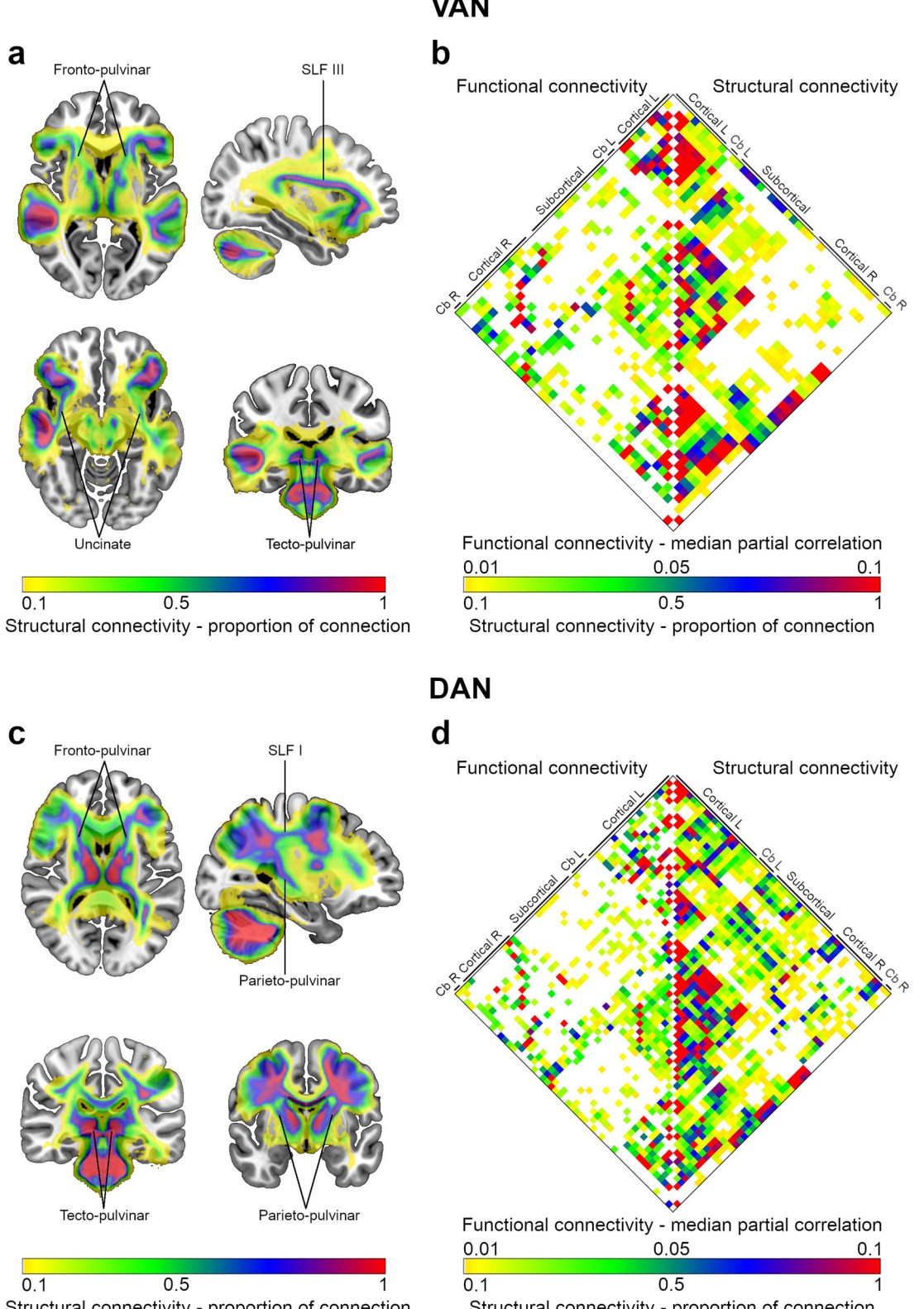

**Fig. 3 Structural and functional connectivity of VAN and DAN nodes. a** Structural connectivity map of the VAN. **b** Matrix with the node-to-node functional and structural connectivity of the VAN, represented on the left and right halves, respectively. **c** Structural connectivity map of the DAN. **d** Matrix with the node-to-node functional and structural connectivity of the DAN, represented on the left and right halves, respectively. Nodes of the matrices were labeled in groups according to their anatomical location. A complete list with node labels is available in Supplementary Tables 1, 2. As indicated, color gradients represent the structural connectivity (expressed as the proportion of connection) or the functional connectivity (defined as the median partial correlation). Cb cerebellum, L left, R right, SLF superior longitudinal fasciculus.

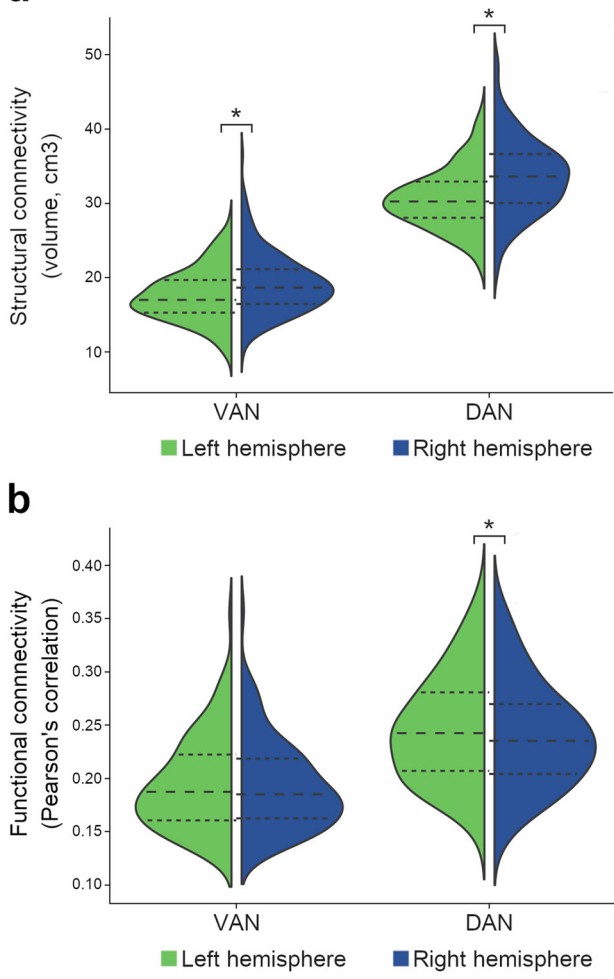

**Fig. 4 VAN and DAN lateralization.** Structural connectivity is expressed in volumes of the structural connection maps (**a**) and functional connectivity in average Pearson's correlations (**b**) across hemispheric nodes. Dashed lines represent the median and the interquartile range; the minimum and maximum correspond to the violin limits. DAN dorsal attention network, VAN ventral attention network. Asterisk (*), significant differences between the right and the left hemispheres ($p < 0.05$; paired analysis; structural connectivity, $n = 177$; functional connectivity, $n = 110$).

The correlation of VAN and DAN brainstem projections with the acetylcholine α4β2 nicotinic receptors was significantly higher in the left hemisphere. In contrast, the correlations with the dopamine and serotonin transporters were higher in the right hemisphere (Fig. 6c).

### Discussion

This study re-examined the VAN and the DAN neuroanatomy by co-registering individual network maps in a common functional space. We propose a comprehensive model of these networks based on the convergence of functional, structural, and neuro-chemical findings. First, we confirmed the initial hypothesis that subcortical structures, namely the pulvinar, the superior colliculi, the head of caudate nuclei, and a group of brainstem nuclei, are constituent elements of the attentional networks. Second, we characterized the structural connections underlying functional connectivity. Deep brain nuclei are densely connected and structural network hubs. Third, we showed that the identified brainstem nuclei projections are spatially correlated with the

acetylcholine α4β2 nicotinic receptors and serotonin and dopamine transporters.

Pulvinar is a high-order thalamic relay nucleus participating in cortical-thalamocortical circuits that modulate information processing[77]. Cytoarchitectonically, the pulvinar is divided into four regions: the anterior pulvinar, the inferior pulvinar, the medial pulvinar, and the lateral pulvinar[78]. The medial pulvinar is particularly important in establishing connections with hetero-modal association areas, such as the superior and inferior temporal, the inferior parietal, the dorsolateral prefrontal, and the orbitofrontal cortices[79]. In our model, the pulvinar regions with the highest statistical level were medial, and we demonstrated that they were structurally connected with VAN cortical areas, through fronto-pulvinar projections, and with DAN cortical areas, by fronto-pulvinar and parieto-pulvinar projections[80–82]. Pulvinar lesions may induce hemispatial neglect[39]. Decades ago, Sprague impressively found that hemispherectomy prompted symptoms of hemispatial neglect in cats which were attenuated by removing the contralesional superior colliculus[83,84]. This effect was later observed in humans[85]. In our model, the pulvinar connects with the superior colliculi through the tecto-pulvinar fibers[86], demonstrating the importance of pulvinar—superior colliculi interactions in attention processes. Therefore, in the context of the so-called Sprague effect, removing the contrale-sional superior colliculus in cats with hemispatial neglect would damage the spared attentional network and might partially compensate for the imbalance in the attentional processing[87,88]. Recently, hemispatial neglect was linked to lesions of the human superior colliculus[40]. The Sprague effect is also mediated by the pedunculopontine nuclei[89,90], which is one of the brainstem nuclei included in our model. The pedunculopontine nuclei possess a population of cholinergic neurons in their caudal portion, giving rise to a distinct network that regulates attentional states and enhances the processing of salient stimuli[91]. The descending projections from these cholinergic neurons innervate the nucleus pontis oralis[92] and the gigantocellular nuclei[93], while their dorsal ascending projections innervate the colliculi[94,95] and several nuclei of the thalamus, including the pulvinar and the mediodorsal nuclei[96]. The pattern of the pedunculopontine projections closely matches the brainstem and thalamic map evidenced in our analysis. Hence, lesion analyses and axonal tracings studies confirm the validity of our subcortical model of the VAN and the DAN.

The graph theory analysis results are consistent with the subcortical nuclei hub role in the VAN and the DAN organization. Centrality measures indicate how connected a node is with other nodes. These measures are considered surrogates of the node's relevance for the flow of information and communication within a network[97,98]. The DAN and the VAN subcortical nuclei had a high degree and betweenness centrality scores, positioning them as networks' core regions as previously suggested[99–101].

The neurotransmitter system correlation analysis reinforced the proposed relationship between the subcortical nuclei of the attention networks. The highest spatial correlation of both networks was with the acetylcholine α4β2 nicotinic receptors. The acetylcholine α4β2 nicotinic receptors have a well-established relationship with sustained attention. Acetylcholine α4β2 nicotinic receptors agonists reduce adult monkey distractibility during matching-to-sample tasks with distractors[102] and increase the firing rate of dorsolateral prefrontal neurons during sustained attention tasks, an effect that is reversed by the co-administration of receptor antagonists[103]. In humans, transdermal nicotine administration improves attentiveness[104,105]. All these observations in animals and humans support the critical role of the subcortical acetylcholinergic system in attentional processes.

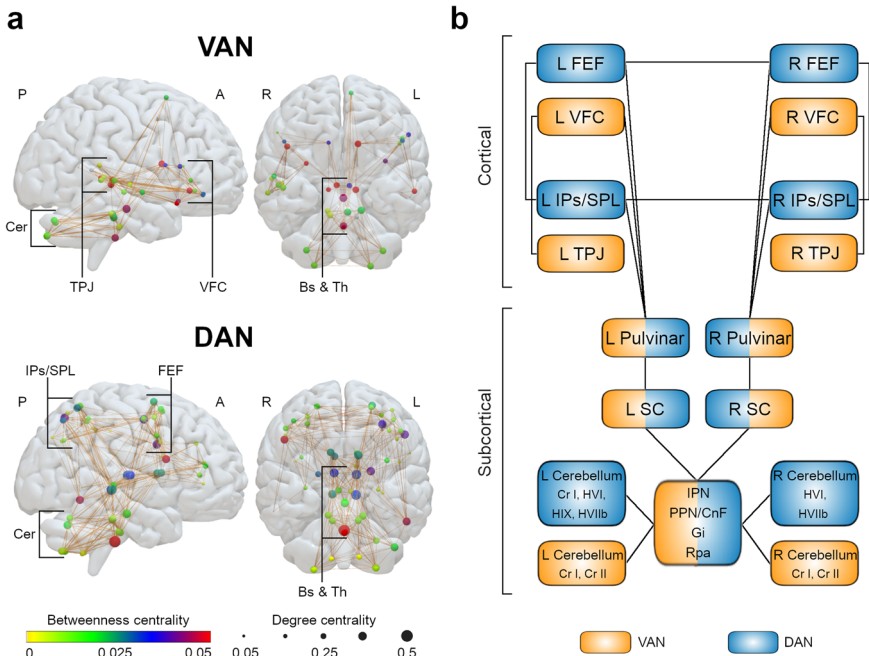

**Fig. 5 Graph theory analysis and anatomical model of the VAN and DAN. a** Graph theory analysis of the VAN and DAN structural connectivity. Circles illustrate nodes. Circle colors represent the median betweenness centrality of each node (according to the color gradient), while circle dimensions represent the median degree centrality. Brown lines represent node-to-node structural connections present in at least half of the subjects. **b** Anatomical model of the VAN and DAN. A anterior, CnF cuneiform nucleus, DAN dorsal attention network, Gi gigantocellular nucleus, IPN interpeduncular nucleus, L left, P posterior, PPN pedunculopontine nucleus, Pul pulvinar, R right, Rpa raphe pallidus nucleus, SC superior colliculus, VAN ventral attention network.

The VAN and DAN brainstem nuclei projections were also spatially correlated with the distribution of dopamine and serotonin transporters. This finding is consistent with the psychopharmacological knowledge about attention. Methylphenidate is the first-line treatment for attention deficit hyperactivity disorder[106]. Pharmacologically, it is a noradrenaline-dopamine reuptake inhibitor with higher potency for dopamine transporters[107,108]. Modafinil is a selective inhibitor of dopamine transporters[109] and produces attention enhancement effects[110,111]. Further studies are needed to understand how the interplay between the nicotinic acetylcholine and the dopamine systems occurs in attention networks, but it might be mediated by their interaction at the levels of the striatum[112,113] and midbrain[114,115]. Serotonin reuptake inhibitors also modulate attentional processes[116]. They increase the perceptual bias towards emotional stimuli[117,118] by regulating the activity of visual processing circuits[116]. Therefore, our improved model of the DAN and VAN functional neuroanatomy appears to reconcile previous neuroimaging and pharmacological findings. As previously suggested[8], additional pharmacological studies will be required to understand the preferential association of VAN with acetylcholine α4β2 nicotinic receptors. Similarly, pharmacological studies are required to shed light on the effect of serotonin transporter on the VAN and to reveal the relationship between dopamine transporters and the DAN. Finally, understanding the relationship between the neurochemical signature and hemispheric functional dominance still requires more research in animals and humans[8].

Characterizing the human brain's subcortical anatomy of attention networks fosters the exploration of a common structural-functional attentional framework across species. Attention is far from being a specific cognitive ability of human beings[119]. Species with either close or distant common ancestors in the phylogenetic tree, such as monkeys, rats, and pigeons, can scan, select and maintain attention to surrounding environmental

stimuli[42,119–121]. A common subcortical attention framework may surpass the challenge of finding the cortical homologs of the human VAN and DAN in other species[43]. Accordingly, future studies might use the subcortical areas we highlighted to explore comparatively the organization of the VAN and the DAN in non-human species.

In our analysis, VAN and DAN structural connectivity maps were right-lateralized. The right lateralization of the VAN is established in the literature. Evidence demonstrates that the SLF III has a larger volume in the right hemisphere and that its anatomical lateralization correlates with visuomotor processing abilities and the asymmetries of visuospatial task performance[7,9,122–125]. The SLF I, the main tract connecting DAN cortical regions, does not show a preferential lateralization[7,9]. However, some DAN areas might be right-lateralized[126]. The right intraparietal sulcus[127] and frontal eye field[128] increase their activity for both visual fields, while the left preferentially reacts to contralateral stimulations. The processing of both visual fields in the right hemisphere is corroborated by right hemisphere stroke patients with hemispatial neglect who also present with deficits in goal-driven selective attention for ipsilateral stimuli[129]. Hence, while the cortical extent of the DAN was not asymmetrical, our structural connectivity analysis, including the cortico-subcortical projection tracts, might have the function-specific dimension of the right lateralization of the DAN.

Regarding functional connectivity, the distribution of VAN was not different between hemispheres, and the DAN was slightly left-lateralized. Task-based fMRI studies indicate right lateralization of the VAN[10,11], but the asymmetry might vary according to the nature of the task[130]. Accordingly, while functional asymmetry is expected for some task-related activations[131], resting-state functional connectivity may not capture function-specific asymmetries due to its global nature.

A limitation of our study is the inability to untangle the different roles and dynamic interactions between the proposed subcortical structures. While the cortical regions of the DAN and

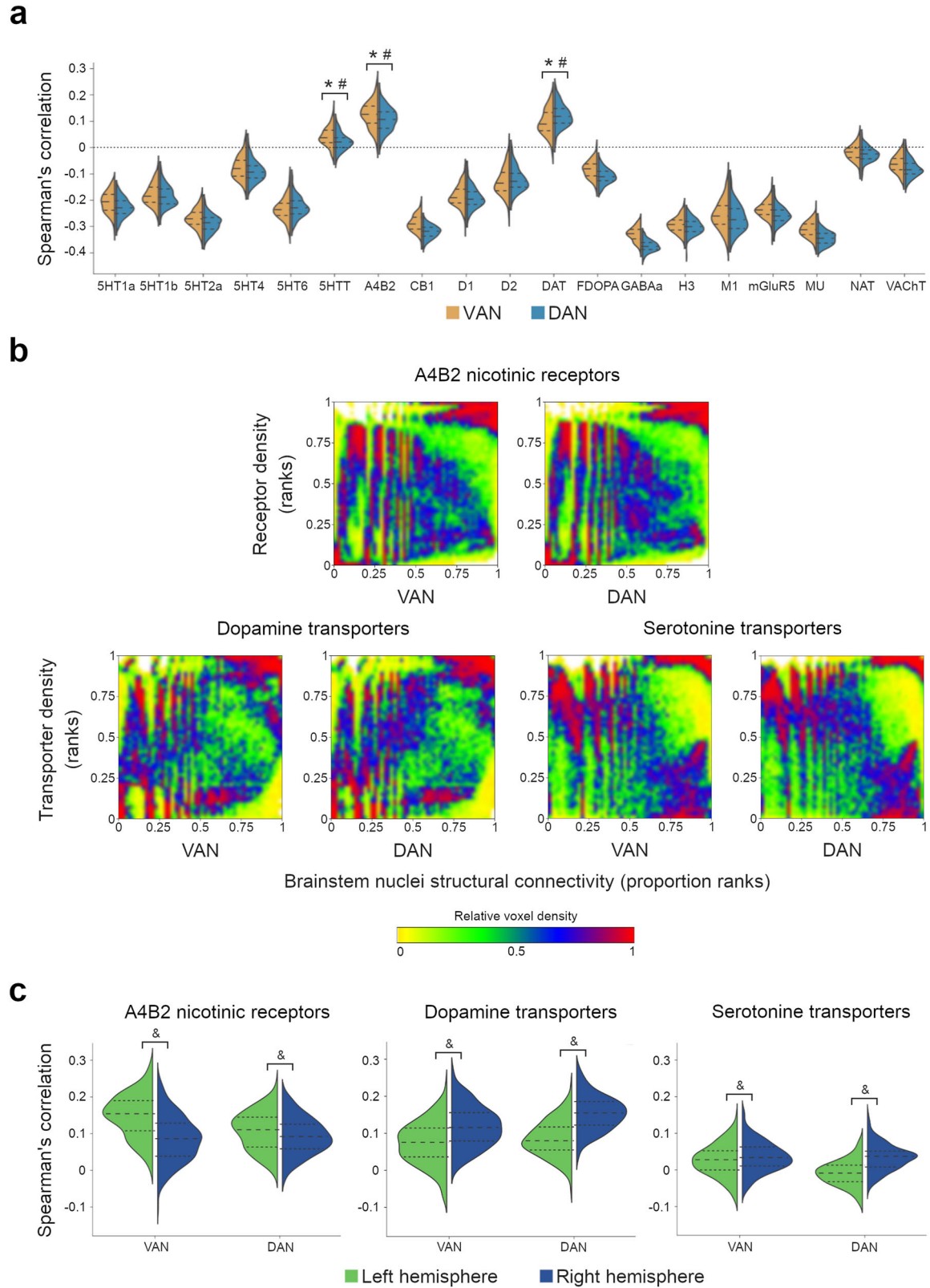

the VAN are quite neatly segregated[132], the subcortical nuclei described in our model probably contributed to both the VAN and the DAN. Future investigations using our model to explore the BOLD signal during task-related *f*MRI in humans or direct electrical recordings in animals might better dissociate the hierarchical organization and functional role of subcortical regions than resting-state *f*MRI. In addition, the neurotransmitter systems normative atlas is derived from different samples[71]. As PET and SPECT tracers are radioactive, it is not possible to map several neurotransmitter systems in the same participants. Although the atlas was replicated in an independent autoradiography dataset and all scans were acquired in healthy volunteers[71], the heterogeneity of the data sources may represent a limitation for its interpretation.

**Fig. 6 Correlation between the structural projections of the brainstem nuclei and the neurotransmitter systems. a** Distributions of the Spearman's correlations for the available maps of neurotransmitter receptors and transporters; for the receptors or transporters with two or more maps available, the mean correlation was calculated. Dashed lines represent the median and the interquartile range; the minimum and maximum correspond to the violin limits. **b** Graphical representation of the statistically significant positive correlations, i.e., the acetylcholine α4β2 nicotinic receptor, dopamine, and serotonin transporter maps. The color map represents the relative voxel density at each graph point. **c** Spearman's correlation of the statistically significant positive correlations with the left and right hemispheres. Dashed lines represent the median and the interquartile range; the minimum and maximum correspond to the violin limits. 5HT1a serotonin 1a receptors, 5HT1b serotonin 1b receptors, 5HT2a serotonin 2a receptors, 5HTT serotonin transporters, A4B2 acetylcholine α4β2 nicotinic receptors, CB1 cannabinoid receptors 1, D1 dopamine receptors 1, D2 dopamine receptors 2, DAT dopamine transporters, FDOPA fluorodopa, GABAa GABAa receptors, H3 histamine receptors 3, M1 muscarinic receptors 1, mGluR5 metabotropic glutamate receptors 5, MU mu-opioid receptors, NAT noradrenaline transporters, VAChT vesicular acetylcholine transporters. * Statistically significant positive correlation, corrected for multiple comparisons ($p < 0.003$); # Statistically significant difference between the VAN and the DAN, corrected for multiple comparisons ($p < 0.017$); & Statistically significant difference between right and left hemispheres, corrected for multiple comparisons ($p < 0.017$); $n = 177$.

In conclusion, this work proposes an improved neuroanatomical model of the VAN and the DAN that includes the pulvinar, the superior colliculi, the head of caudate nuclei, and a group of brainstem nuclei interrelated with the acetylcholine nicotinic and the dopamine and serotonin transporter systems. This comprehensive framework reconciles behavioral, electrophysiological, and psychopharmacological data and provides a shared foundation to explore the neural basis of attention across different species and brain pathologies.

## Methods

**Resting-state functional imaging (rs-fMRI).** We used 110 7 T resting-state functional MRI datasets from the Human Connectome Project S1200[133]. Images were preprocessed and registered to the MNI152 space as specified in the Human Connectome Project protocol (http://www.humanconnectome.org/storage/app/media/documentation/s1200/HCP_S1200_Release_Reference_Manual.pdf; Glasser et al. 2013). The Human Connectome Project open access data use terms were followed.

**VAN and DAN maps in the structural space.** VAN and DAN maps were computed using seed regions of interest defined in the functional cortical parcellation map[14]. This template includes 23 VAN parcels (11 in the left and 12 in the right hemisphere) and 32 DAN parcels (19 in the left and 13 in the right hemisphere). This parcellation was performed according to resting-state functional connectivity patterns. Each parcel has a homogeneous resting-state functional connectivity signature and is separated from neighboring parcels by abrupt changes in their connectivity profile[14].

We calculated functional correlation maps seeded from each VAN cortical parcel using the Funcon-Connectivity tool implemented in the Brain Connectivity and Behavior toolkit (http://toolkit.bcblab.com)[134]. This tool computes Pearson's correlation between a seed region's mean resting-state activity and the brain's other voxels. Then, the median of the 23 functional connectivity maps (generated from the 23 VAN seeds) was computed to obtain the VAN's most representative map for each participant. We chose a median because it is less affected by outliers than the mean[135]. 110 individual VAN maps in the MNI152 were obtained (i.e., one per subject). The same steps were performed to obtain 32 DAN maps.

**VAN and DAN maps in the functional space.** The 110 individual VAN Pearson's correlation maps in the MNI152 space were aligned in a functional space to optimize their interindividual alignment of functional areas[50–53]. We used the Advanced Normalization Tools (ANTs) script "buildtemplateparallel.sh" to perform an iterative ($n = 4$) diffeomorphic transformation to a common space[62,136]. Cross-correlation was set as the similarity measure and greedy SyN as the transformation model[137,138]. The resulting transformation warps were applied to the MNI152 aligned VAN maps, using the ANTs' script "WarpImageMultiTransform" to represent the 110 individual VAN maps in the functional space. The same steps were performed with the 110 DAN Pearson's correlations maps. A schematic representation of the functional alignment steps is available in Supplementary Fig. 1.

To calculate group statistical VAN and DAN maps, we performed a permutation inference analysis using FSL's "randomise" one-sample (5000 permutations) and applied a threshold-free cluster enhancement[139]. To evaluate the similarity between the VAN and DAN statistical maps, the t-maps were z-transformed, and a conjunction analysis was computed[140]. A difference map was also calculated by subtracting the median DAN Pearson's correlation map from the VAN. Illustrations were produced in SurfIce (https://www.nitrc.org/projects/surfice/) and MRIcroGL (https://www.nitrc.org/projects/mricrogl/).

**Anatomical validation of the subcortical structures.** To identify thalamic nuclei, we visually compared our results with the DISTAL (Deep brain stimulation Intrinsic Template Atlas; Ewert et al. 2018) and the THOMAS (Thalamus Optimized Multi Atlas Segmentation; Su et al. 2019) atlases. The DISTAL atlas is a high-resolution template of subcortical structures in the MNI space used as a reference to localize targets for deep brain stimulation[141]. The DISTAL atlas segmentation was performed manually, based on histology, structural imaging, and diffusion-weighted imaging[141,143]. The THOMAS atlas is a template of thalamic nuclei derived from the manual segmentation of 20 White-Matter-Nulled Magnetization Prepared Rapid Gradient Echo (MP-RAGE) 7 T datasets warped to the MNI space[142]. We used the WIKIBrainStem atlas to identify the brainstem nuclei[144]. This template is based on mesoscopic T2-weighted and diffusion-weighted images obtained from the ultra-high-field scanning (11.7 T) of an ex vivo human specimen. It provides detailed segmentations of 99 brainstem structures[144].

**Tractography analysis.** We analyzed the structural connectivity of the VAN and DAN, including the new subcortical structures identified in our resting-state functional connectivity analysis. Tractography was computed using 177 diffusion-weighted images from the 7 T dataset of the Human Connectome Project[145]. The scanning parameters are detailed in Vu et al.[145]. Preprocessing was performed according to the default Human Connectome Project pipeline (v3.19.0)[133]. The Human Connectome Project open access data use terms were followed. Tractography processing was prepared as described in Thiebaut de Schotten et al.[67] (available at http://opendata.bcblab.com). Briefly, a whole-brain deterministic algorithm was employed using StarTrack (https://mr-startrack.com), applying a damped Richardson-Lucy algorithm optimized for spherical deconvolution[146]. Then, the individual whole-brain streamline tractograms were registered to the MNI152 space. First, they were converted into density maps, in which the voxel densities corresponded to the number of streamlines crossing each voxel[67]. Second, individual density maps were aligned to a standard template using the Greedy symmetric diffeomorphic normalization of the Advanced Normalization Tools pipeline[136]. Third, the resulting template was co-registered to the MNI152 2 mm template using the FSL's tool "flirt"[147]. Finally, the resulting transformation warps were applied to the individual whole-brain streamline tractography using Tract Querier[148].

Then, we computed the structural connectome of the VAN and DAN models. The cortical nodes were defined according to Gordon et al.[14]. To determine the subcortical regions of interest, we selected the statistically significant voxels of the subcortical structures identified in the previous sections with a median Pearson's correlation above $r = 0.1$. This correlation threshold was applied to avoid including voxels significantly associated with the network but with weak correlations[149]. The streamlines that crossed at least two ROIs (cortico-cortical, cortico-subcortical, or subcortical-subcortical) were selected using the MRtrix3's tool "tckedit"[150]. Afterward, the selected streamlines were converted into streamline density maps using the MRtrix3's tool 'tckmap'[150]. The streamline density maps were binarized, and a group-level overlap map was computed.

**ROI-to-ROI structural and functional connectivity analysis.** We used MRtrix3's tool "tck2connectome" to analyze ROI-to-ROI structural connectivity. The cortical and subcortical ROIs were defined as stated in the previous section. Regarding ROI-to-ROI functional connectivity, we computed the partial correlation between the network nodes using the nilearn's function "ConnectivityMeasure"[151]. The illustrations of the connectivity matrices were created with Matplotlib 3.4.2[152].

**Networks lateralization.** We assessed the lateralization of the VAN and DAN networks. For functional connectivity, the average Pearson's correlation across each hemisphere's VAN and DAN nodes was calculated using the FSL's function "fslmeants". The obtained values were compared between the right and left hemispheres. For structural connectivity, we extracted the fiber tracts that crossed two nodes of the same hemisphere. Then, the fiber tracts were converted into volume maps using the MRtrix3's tool "tckmap", and the individual volumes were

compared between the two hemispheres[150]. Data were presented as mean (with standard deviations) or median (with interquartile ranges), and paired analyses were performed with paired *t*-test or Wilcoxon test, according to their distribution.

**Graph theory analysis of structural connectivity**. To analyze if the newly identified subcortical nuclei would be core regions in the networks, we performed a graph theory analysis of the hub properties of the VAN and DAN nodes. Two measures were used, the degree centrality and the betweenness centrality[97]. Degree centrality denotes the fraction of nodes connected to the node of interest. Betweenness centrality is the fraction of all-pairs shortest paths that pass through the node of interest[97]. In graph theory, nodes with high centrality are considered network hubs, i.e., they play a crucial role in the global network function[153].

The 177 individual binarized structural connectivity matrices were converted into undirected connectivity graphs, and both measures were calculated using the NetworkX package (https://networkx.org/). ROIs, as defined in the previous sections, constituted the network nodes. The streamlines that crossed at least two ROIs defined network vertices. Considering the conservative parameters of our tractography adjusted over the years to match post-mortem Klingler dissections[154–157], there was no threshold for the streamline considered for binarization. Additionally, the streamline count does not accurately reflect the number of axonal projections between regions or the strength of connectivity[99,158], and previous work showed that the overall results of the network analysis do not change with modifications in the streamline count binarization threshold[159]. Then, we calculated the median value of both measures across the 177 network graphs for each node. The illustrations of the network graphs were created with SurfIce (https://www.nitrc.org/projects/surfice/).

**Structural correlations with the neurotransmitter system**. We studied the relationship between the proposed neuroanatomical models' subcortical structural projections and the neurotransmitter systems' spatial distribution. First, we selected the newly identified brainstem nuclei that synthesize neurotransmitters, according to the cytochemical evidence in the literature. Second, we computed the structural projections of these nuclei to the remaining nodes of the VAN and DAN, i.e., we selected the streamlines that crossed the brainstem nuclei of interest and every other node of the network, using the MRtrix3 tool "tckedit"[150]. Then, we used the MRtrix3 tool "tckmap" to map those streamlines into the MNI space[150] and computed the individual Spearman's correlation between the spatial distribution of the created structural projection map and the neurotransmitter maps provided by Hansen and colleagues using the neuromaps' tool "compare_images"[71,160]; https://netneurolab.github.io/neuromaps/). We obtained the correlation values distribution between the 110 individual VAN and DAN maps and each neurotransmitter map. To analyze if the obtained distributions (each composed of 110 correlation values) were significantly higher than zero, a non-parametric statistical test was performed (one-sided Wilcoxon test). The obtained *p* values were corrected for multiple comparisons using the Bonferroni correction. Finally, we analyzed whether the correlation distributions were different between VAN and DAN, and if they were different between hemispheres (paired *t*-test or Wilcoxon test, according to data distribution; the Bonferroni correction was also applied). A supplemental pairwise analysis was performed. The average map of the 110 individual VAN and DAN structural projection maps was correlated with the neurotransmitter maps (Spearman's correlation; neuromaps' tool "compare_images"; https:// netneurolab.github.io/neuromaps/)[160]. To control for spatial autocorrelations and reduce the risk of false positive results, statistical significance was inferred based on null models generation[161–163]. Volumetric data were parcellated according to the Automated Anatomical Labeling atlas 3 (AAL3; Rolls et al.[164]), using the neuromaps' utility "Parcellater" (https://netneurolab.github.io/neuromaps/)[160]. AAL3 was chosen because it includes cortical and subcortical parcels. The null parcellations were generated from the average VAN and DAN structural projection maps using the neuromaps' function "nulls.-burt2020" (5000 permutations, generating 5000 null parcellations; https://netneurolab. github.io/neuromaps/)[160,162]. The graphical representations were created with Matplotlib 3.4.2 and Datashader 0.13.0 (Hunter[152]; https://datashader.org).

**Reporting summary**. Further information on research design is available in the Nature Portfolio Reporting Summary linked to this article.

## Data availability
The presented brain maps are openly available at https://neurovault.org/collections/ XONZLGPJ/, the resting-state functional 7 T MRI datasets in the Human Connectome Project S1200 dataset, and the processed tractographies at http://opendata.bcblab.com. Data were provided by the McDonnell Center for Systems Neuroscience at Washington University. All other data were available from the corresponding author on reasonable request.

## Code availability
Analyses were conducted using open software and toolboxes, as specified in the methods. The Funcon-Connectivity code is openly available at https://github.com/chrisfoulon/BCBToolKit; the ANTs scripts "buildtemplateparallel.sh" and "WarpImageMultiTransform" at https:// github.com/ANTsX/ANTs; the code where these scripts were applied for the functional

alignment performed in this work at https://github.com/Pedro-N-Alves/VAN_DAN_ functional_alignment (doi: 10.5281/zenodo.7307027); the "easythresh_conj.sh" code (used for the conjunction analysis) at https://warwick.ac.uk/fac/sci/statistics/staff/academic-research/ nichols/; the "tckedit", "tckmap", and "tck2connectome" commands' codes at https://github. com/MRtrix3/mrtrix3; the "ConnectivityMeasure" script at https://github.com/nilearn/ nilearn; the "betweenness_centrality" and "degree_centrality" scripts at https://github.com/ networkx/networkx; and the "compare_images", "Parcellater", and "nulls.burt202" scripts at https://github.com/netneurolab/neuromaps.

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

## Acknowledgements
This project has received funding from the European Research Council (ERC) under the European Union's Horizon 2020 research and innovation program (grant agreement No. 818521; M.T.d.S.), the Marie Skłodowska-Curie program (grant agreement No. 101028551; S.J.F.), a Donders Mohrmann Fellowship (S.J.F., No. 2401512), "Prémio João Lobo Antunes 2018"—SCML (P.N.A.), and "Bolsa de Investigação em Doenças Vasculares Cerebrais 2017"—SPAVC (P.N.A.). Additional funding comes from the University of Bordeaux's IdEx "Investments for the Future" program RRI "IMPACT," which received financial support from the French government. M.C. was supported by MIUR— Departments of Excellence Italian Ministry of Research (MART_ECCELLENZA18_01); Fondazione Cassa di Risparmio di Padova e Rovigo (CARIPARO)(Grant Agreement number 55403); Ministry of Health Italy NEUROCONN (RF-2008 -12366899); Celeghin Foundation Padova (CUP C94I20000420007); BIAL foundation grant (No. 361/18); H2020 European School of Network Neuroscience-euSNN, H2020-SC5-2019-2, (Grant Agreement number 869505); H2020 VARCITIES, H2020-SC5-2019-2 (Grant Agreement number 869505); Ministry of Health Italy: EYEMOVINSTROKE (RF-2019-12369300).

## Author contributions

P.N.A. implemented part of the methods, performed the analyses, contributed conceptually, and wrote the manuscript. S.J.F. and M.C. contributed conceptually and edited the paper. M.T.d.S. conceived and coordinated the study, reviewed the neuroimaging data, wrote the manuscript, and provided funding.

## Competing interests
Michel Thiebaut de Schotten is an Editorial Board Member for *Communications Biology*, but was not involved in the editorial review of, nor the decision to publish this article. The authors declare no competing interests.
