## [Peer Review File · Communications Biology]

Reviewers' comments:

Reviewer #1 (Remarks to the Author):

In the manuscript entitled "The subcortical and neurochemical organization of the ventral and dorsal attention networks," the authors map the subcortical architecture of the ventral and dorsal attention networks, reveal the structural network hub organization of the corresponding subcortical nuclei, and establish their interrelations with the neurotransmitter systems. This study provides valuable resources to researchers who study the attention networks in the brain. The paper is well written and structured. I only have several minor comments on the current version of the manuscript.

1. The authors performed the graph theory analysis, but I did not find any detailed discussion about this in the manuscript. Is there anything else the authors can discuss other than just describing that the subcortical nuclei are densely connected structural hubs?
2. Only in Fig. 5a, the authors present results of DAN first and results of VAN second. Would it be better to switch this order to make it consistent with all other figures?
3. I would suggest to add descriptions of Fig. 6b to the main text of Results (currently, these are missing).
4. Line 591: Should "[INSERT CORBETTA 2008]" read "(Corbetta et al. 2008),"

Reviewer #2 (Remarks to the Author):

This study argues that subcortical structures play a crucial role in neural mechanisms of attention. As a consequence, the neuroanatomical framework of attentional mechanisms should extend well beyond a cortico centric model. This paper proposes a new framework for a comprehensive understanding of attention across species and disorders by combining functional, structural, and neurochemical evidence.

In their paper, Alves et al. test the hypothesis that basal ganglia and brainstem nuclei, including the pulvinar, striatum, superior colliculi, and locus coeruleus, provide the phylogenetically relevant and functional component of the attention network. Using functional alignment methods applied to functional connectivity analyses of resting-state fMRI, the authors map the subcortical architecture of Ventral and Dorsal Attention Networks (VAN and DAN). For this purpose, VAN and DAN maps in the structural and functional space were obtained and the structural and functional connectivity of the VAN and DAN nodes was analyzed. Additionally, the authors conducted lateralization assessments, graph theoretical analyses, and neurotransmitter correlations.

The authors found that the subcortical structures are essential elements of the attentional network. They conclude this based on results revealing that the deep brain nuclei are densely connected and structural network hubs. Further, the work also established a spatial correlation between brainstem nuclei projections and acetylcholine $\alpha 4\beta 2$ nicotinic receptors and serotonin and dopamine transporters. The authors conclude by proposing an improved neuroanatomical model of the VAN and the DAN, which incorporates the pulvinar nucleus, the superior colliculi, the caudate head nuclei, and groups of neurons in the brain stem associated with the acetylcholine nicotinic system and dopamine and serotonin transporters. The authors further claim that this new framework combines behavioral, electrophysiological, and psychopharmacological data and provides a shared basis for exploring the neural basis of attention in a wide range of species and brain diseases.

This paper's title and abstract are appropriate to its content. The article is also well constructed, the methods are appropriate and the analysis was well done. One of the main strengths of this paper is

that it addresses an interesting hypothesis, provides an innovative framework based on combining modalities, and provides a novel insight. The article represents an excellent neuroscientific study that will likely influence our understanding of the attentional network. Another strength of the paper is the open availability of the presented brain maps.

Suggested revisions:

1) A suggestion for revision is to provide open access to the scripts as well, rather than requiring requests.

2) One of the major findings of the paper is that "Deep brain nuclei are densely connected and structural network hubs." However, the methods does not describe how this is determined which nodes are considered hubs in the network. More details are needed about how this is determined. In general, the description of the graph analysis is not very detailed and more details about for instance whether thresholding was applied or not would be of interest for the reader.

3) It was considered that VAN and DAN exhibited right-lateralization structurally, while DAN showed a slight left-lateralization functionally. A structural lateralization finding is much more surprising and interesting than functional lateralization in my opinion, while only functional lateralization is addressed in the discussion of VAN and DAN. I would like to see discussion of why VAN and DAN might be right lateralized.

4) Previsouly line 590
Inser CORBETTA 2008 line 591
Foundation page 631

Reviewer #3 (Remarks to the Author):

The authors used functional alignment applied to rsFC to map the subcortical architecture of the VAN and DAN. They also studied associations to diffusion MRI connectivity and neurochemical transmitter systems.

This is a overall a nicely written and motivated study. I only have few comments, mostly pertaining to providing deeper motivation for some of the analyses, clarification of the network taxonomy, and to clarify the spatial associations with neurotransmitter systems and to acknowledge limitations of such a cross-dataset contextualization. Please, find my specific comments below.

1) Introduction: Please clarify in how far the taxonomy of networks is comparable to eg Yeo et al. 2011 with respect to DAN and VAN/Saliience networks. In the introduction, for example, the authors mention that VAN and DAN are both fronto-parietal networks. While this is of course also true, i would recommend to drop fronto-parietal here to differentiate VAN and DAN from the fronto-patietal network (FPN) as defined by resting-state fMRI (see eg Yeo et al 2011 J Neurophysiol; sometimes also referred to as central executive network).

2) Introduction. When talking about functional alignment, further motivation, explanation and contextualization of the technique is needed. I would suggest to also differentiate the applied technique from eg 'hyperalignment' approaches for task based data, and to discuss similarities and differences from the apprach in the work from alignments working on eg resting state connectivity patterns / 'gradients' (eg. Nenning et al., Xu et al.).

3) Introduction. Please motivate diffusion connectivity analysis in the introduction as well, as this seems to be one of the methods used in the study of subcortical networks. Ditto for the neurochemical/transmitter resources.

4) Methods: I am not fully clear on how the functional alignment was done in the current work. The authors mention that VAN/DAN maps were aligned with ANTS based on cross-correlation similarity, but I fail to understand the specifics here and which functional features were specifically used for alignment (connectivity patterns, time series, etc). Further details, and potentially a schematic supplementary figure, would greatly help.

5) The lateralization analysis was interesting, but came a bit out of the blue. I would add further rationale in the introduction as well on why one should inspect inter-hemispheric asymmetries in the context of these networks and their subcortical components.

6) Results: Figure 1 could benefit from medial cortical views as well.

7) Results. The neurotransmitter associations are interesting, but it needs to be highlighted that these maps come from entirely different datasets with likely different socio-demographic criteria as HCP. This should be highlighted in abstract and other parts for the manuscripts, and limitations/indirectness of the approach should be acknowledged.

8) Please clarify how spatial similarity between neurotransmitter and imaging maps was tested. Did the authors use null models that control for autocorrelation to establish significances?

We would like to thank the reviewers for their insightful comments.

Below, we address point by point all comments. The corresponding changes in the main text are highlighted in cyan.

Reviewer #1

In the manuscript entitled "The subcortical and neurochemical organization of the ventral and dorsal attention networks," the authors map the subcortical architecture of the ventral and dorsal attention networks, reveal the structural network hub organization of the corresponding subcortical nuclei, and establish their interrelations with the neurotransmitter systems. This study provides valuable resources to researchers who study the attention networks in the brain. The paper is well written and structured. I only have several minor comments on the current version of the manuscript.

R: We thank the reviewer for their very positive feedback on our manuscript.

1) The authors performed the graph theory analysis, but I did not find any detailed discussion about this in the manuscript. Is there anything else the authors can discuss other than just describing that the subcortical nuclei are densely connected structural hubs?

R: We thank the reviewer for the opportunity to expand. A new paragraph was introduced in the discussion about the interpretation of the graph theory analysis results.

- Discussion:

Added: *“The graph theory analysis results are consistent with the subcortical nuclei hub role in the VAN and DAN organization. Centrality measures indicate how connected a node is with other nodes. These measures are considered surrogates of the node’s relevance for the flow of information and communication within a network (Girvan and Newman 2002; Bullmore and Sporns 2009). DAN and VAN subcortical nuclei had a high degree and betweenness centrality scores, positioning them as networks’ core regions as previously suggested (Barabási and Albert 1999; Hagmann et al. 2008; Gong et al. 2009).”*

2) Only in Fig. 5a, the authors present results of DAN first and results of VAN second. Would it be better to switch this order to make it consistent with all other figures?

R: We corrected this inconsistency and uploaded a new version of figure 5 where the VAN appears first and the DAN second.

3) I would suggest to add descriptions of Fig. 6b to the main text of Results (currently, these are missing).

R: As suggested, the following description was added to the Results:

- Results, Correlation with the neurotransmitter system

Added: *“The scatterplots representing the distributions of the significantly correlated systems are presented in figure 6b.”*

4) Line 591: Should “[INSERT CORBETTA 2008]” read “(Corbetta et al. 2008),”?

R: Thank you. The reference (Corbetta et al. 2008) has now been properly inserted.

Reviewer #2

This study argues that subcortical structures play a crucial role in neural mechanisms of attention. As a consequence, the neuroanatomical framework of attentional mechanisms should extend well beyond a cortico centric model. This paper proposes a new framework for a comprehensive understanding of attention across species and disorders by combining functional, structural, and neurochemical evidence.

In their paper, Alves et al. test the hypothesis that basal ganglia and brainstem nuclei, including the pulvinar, striatum, superior colliculi, and locus coeruleus, provide the phylogenetically relevant and functional component of the attention network. Using functional alignment methods applied to functional connectivity analyses of resting-state fMRI, the authors map the subcortical architecture of Ventral and Dorsal Attention Networks (VAN and DAN). For this purpose, VAN and DAN maps in the structural and functional space were obtained and the structural and functional connectivity of the VAN and DAN nodes was analyzed. Additionally, the authors conducted lateralization assessments, graph theoretical analyses, and neurotransmitter correlations.

The authors found that the subcortical structures are essential elements of the attentional network. They conclude this based on results revealing that the deep brain nuclei are densely connected and structural network hubs. Further, the work also established a spatial correlation between brainstem nuclei projections and acetylcholine $\alpha4\beta2$ nicotinic receptors and serotonin and dopamine transporters. The authors conclude by proposing an improved neuroanatomical

model of the VAN and the DAN, which incorporates the pulvinar nucleus, the superior colliculi, the caudate head nuclei, and groups of neurons in the brain stem associated with the acetylcholine nicotinic system and dopamine and serotonin transporters. The authors further claim that this new framework combines behavioral, electrophysiological, and psychopharmacological data and provides a shared basis for exploring the neural basis of attention in a wide range of species and brain diseases.

This paper's title and abstract are appropriate to its content. The article is also well constructed, the methods are appropriate and the analysis was well done. One of the main strengths of this paper is that it addresses an interesting hypothesis, provides an innovative framework based on combining modalities, and provides a novel insight. The article represents an excellent neuroscientific study that will likely influence our understanding of the attentional network. Another strength of the paper is the open availability of the presented brain maps.

R: We thank the reviewer for their very positive feedback.

1) A suggestion for revision is to provide open access to the scripts as well, rather than requiring requests.

R: We thank the reviewer for this suggestion. The open access links of the scripts were added.

- Code availability statement

Added: *“The Funcon-Connectivity code is openly available at <https://github.com/chrisfoulon/BCBToolKit>; the ANTs scripts ‘buildtemplateparallel.sh’ and ‘WarpImageMultiTransform’ at*

https://github.com/ANTsX/ANTs; the code where these scripts were applied for the functional alignment performed in this work at https://github.com/Pedro-N-Alves/VAN_DAN_functional_alignment; the 'easythresh_conj.sh' code (used for the conjunction analysis) at https://warwick.ac.uk/fac/sci/statistics/staff/academic-research/nichols/; the 'tckedit', 'tckmap' and 'tck2connectome' commands' codes at https://github.com/MRtrix3/mrtrix3; the 'ConnectivityMeasure' script at https://github.com/nilearn/nilearn/; the 'betweenness centrality' and 'degree centrality' scripts at https://github.com/networkx/networkx; and the 'compare_images', 'Parcellater' and 'nulls.burt202' scripts at https://github.com/netneurolab/neuromaps."

2) One of the major findings of the paper is that “Deep brain nuclei are densely connected and structural network hubs.” However, the methods does not describe how this is determined which nodes are considered hubs in the network. More details are needed about how this is determined. In general, the description of the graph analysis is not very detailed and more details about for instance whether thresholding was applied or not would be of interest for the reader.

R: We extended the explanation about the graph theory analysis in the methods to clarify how the centrality measures were calculated. Specifically, we provided details about how the threshold was performed and its rationale.

- Methods, Graph theory analysis of structural connectivity

Added: *“ROIs, as defined in the previous sections, constituted the network nodes. The streamlines that crossed at least two ROIs defined network vertices. Considering the conservative parameters of our tractography adjusted over the years to match post-mortem Klingler dissections (Thiebaut de Schotten et al. 2011; Catani et al.*

2012; Vergani et al. 2014; Catani 2019), there was no threshold for the streamline considered for binarization. Additionally, streamline count does not accurately reflect the number of axonal projections between regions or the strength of connectivity (Gong et al. 2009; Jones et al. 2013), and previous work showed that the overall results of the network analysis do not change with modifications in the streamline count binarization threshold (Shu et al. 2011).”

In addition, we further explain the relationship between these measures and the classification of a node as a network hub.

- Methods, Graph theory analysis of structural connectivity

Added: “*In graph theory, nodes with high centrality are considered network hubs, i.e., they play a crucial role in the global network function (van den Heuvel and Sporns 2013).*”

Finally, a new paragraph was added to the discussion about interpreting graph theory analysis results.

- Discussion

Added: “*The graph theory analysis results are consistent with the subcortical nuclei hub role in the VAN and DAN organization. Centrality measures indicate how much a node is connected with other nodes. They are considered surrogates of the node’s relevance over the flow of information and communication in the network (Girvan and Newman 2002; Bullmore and Sporns 2009). DAN and VAN subcortical nuclei had high degree and betweenness centrality scores, positioning them as networks’ core regions as previously suggested (Barabási and Albert 1999; Hagmann et al. 2008; Gong et al. 2009).*”

3) It was considered that VAN and DAN exhibited right-lateralization structurally, while DAN showed a slight left-lateralization functionally. A structural lateralization finding is much more surprising and interesting than functional lateralization in my opinion, while only functional lateralization is addressed in the discussion of VAN and DAN. I would like to see discussion of why VAN and DAN might be right lateralized.

R: We thank the reviewer for this suggestion. We expanded the discussion on the putative reasons behind these results.

- Discussion

Added (after “In our analysis, VAN and DAN structural connectivity maps were right-lateralized.”): *“The right lateralization of the VAN is established in the literature. Evidence demonstrates that the SLF III has a larger volume in the right hemisphere and that its anatomical lateralization correlates with visuomotor processing abilities and the asymmetries of visuospatial task performance (Thiebaut de Schotten et al. 2011a; Chechlacz et al. 2015; Budisavljevic et al. 2017; Cazzoli and Chechlacz 2017; Howells et al. 2018; Amemiya et al. 2021). The SLF I, the main tract connecting DAN cortical regions, does not show a preferential lateralization (Thiebaut de Schotten et al. 2011; Amemiya et al. 2021). However, some DAN parts might be right-lateralized (Bartolomeo and Seidel Malkinson 2019). The right intraparietal sulcus (Sheremata and Silver 2015) and frontal eye field (Szczepanski et al. 2010) increase their activity for both visual fields, while the left preferentially reacts to contralateral stimulations. The processing of both visual fields in the right hemisphere is corroborated by right hemisphere stroke patients with hemispatial neglect who also present with deficits in goal-driven selective attention for ipsilateral*

stimuli (Snow and Mattingley 2006). Hence, while the cortical extent of the DAN was not asymmetrical, our structural connectivity analysis, including the cortico-subcortical projection tracts, might have the function-specific dimension of the right-lateralization of the DAN.”

4) Previsouly line 590

Inser CORBETTA 2008 line 591

Fundation page 631

R: Thank you for spotting these typos. The reference (Corbetta et al. 2008) was now properly inserted and the typos corrected.

Reviewer #3

The authors used functional alignment applied to rsFC to map the subcortical architecture of the VAN and DAN. They also studied associations to diffusion MRI connectivity and neurochemical transmitter systems.

This is a overall a nicely written and motivated study. I only have few comments, mostly pertaining to providing deeper motivation for some of the analyses, clarification of the network taxonomy, and to clarify the spatial associations with neurotransmitter systems and to acknowledge limitations of such a cross-dataset contextualization. Please, find my specific comments below.

R: We thank the reviewer for their very positive feedback.

1) Introduction: Please clarify in how far the taxonomy of networks is comparable to eg Yeo et al. 2011 with respect to DAN and VAN/Saliience

networks. In the introduction, for example, the authors mention that VAN and DAN are both fronto-parietal networks. While this is of course also true, i would recommend to drop fronto-parietal here to differentiate VAN and DAN from the fronto-patietal network (FPN) as defined by resting-state fMRI (see eg Yeo et al 2011 J Neurophysiol; sometimes also referred to as central executive network).

R: The taxonomy used in this work is comparable to Yeo et al. 2011, regarding DAN, VAN and the fronto-parietal network. DAN and VAN were the focus of this work and are associated with discrete components of the attentional processes – DAN is goal-directed and VAN a stimulus-driven system. The fronto-parietal network is an executive system related to decision-making and task-control, namely task initiation, maintenance, and monitoring (Dosenbach et al. 2007; Vincent et al. 2008). Although some works argue that the fronto-parietal network might include the DAN (Dosenbach et al. 2007), the functional distinction between VAN, DAN the fronto-parietal network is defined in Yeo et al. 2011, in the Gordon et al. 2016 (the parcellation used in this work) and used in other widely accepted functional parcellations (Power et al. 2011, Schaefer et al. 2018).

We agree that the use of ‘fronto-parietal’ here may be ambiguous. The term was removed:

- Introduction

Amended: *“The Dorsal Attention Network (DAN) encodes and maintains preparatory signals and modulates top-down sensory (visual, auditory, somatosensory) regions.”*

Amended: *“In contrast, the Ventral Attention Network (VAN) is recruited when attention is re-oriented to novel behaviorally relevant events.”*

In addition, we added the references Yeo et al. 2011, Power et al. 2011 and Schaefer et al. 2018 to the introduction, and mentioned the issue of heterogeneous taxonomies:

- Introduction:

Added: *“Thanks to this synchronization, the two networks have consistently been identified and segregated in resting-state fMRI cortical parcellations (Power et al. 2011; Yeo et al. 2011; Gordon et al. 2016; Schaefer et al. 2018), although their taxonomy has not always been homogenous in the literature (Eickhoff et al. 2018; Uddin et al. 2019).”*

2) Introduction. When talking about functional alignment, further motivation, explanation and contextualization of the technique is needed. I would suggest to also differentiate the applied technique from eg 'hyperalignment' approaches for task based data, and to discuss similarities and differences from the approach in the work from alignments working on eg resting state connectivity patterns / 'gradients' (eg. Nenning et al., Xu et al.).

R: We thank the reviewer for this suggestion. We expanded the explanations about the alignment methods. First, we detailed the problems associated with brain alignment exclusively based on structural features. Second, we discuss the methodological developments in surface-based functional alignment of functional data to overcome these problems referring to the suggested articles by Nenning et al. 2020 and Xu et al. 2020. Third, the concept of hyperalignment was presented to differentiate it from the methodology used in this work. Finally, we explained the concept behind the volumetric-based alignment of functional networks that we performed and the optimizations it can bring to deep brain nuclei mapping.

The following changes were made:

- Introduction

Added (after “*In contrast, advanced methods of functional alignment improve structural-functional correspondence across subjects*”): “*Further, surface interindividual alignment based on morphological features, such as cortical folding, fairly aligns unimodal cortical areas, such as the primary visual and motor cortices, but poorly overlaps higher-order cortical areas (Fischl et al. 2008; Mueller et al. 2013). Methods of functional alignment based on fMRI signals during cognitive activation paradigms (Sabuncu et al. 2010; Conroy et al. 2013) and resting-state fMRI connectivity patterns (Langs et al. 2015; Nenning et al. 2020) provided better function matching and have also been used for cross-species functional comparisons (Xu et al. 2020). Functional alignment is different from hyperalignment techniques that project shared neural information beyond the three-dimensional anatomical space, i.e., in high-dimensional spaces (Haxby et al. 2011, 2020; Guntupalli et al. 2016)*”

Amended: “*At the subcortical level, our team also demonstrated that functional alignment methods can optimize the group-level mapping of functional networks, improving functional correlations and uncovering a network’s deep brain nuclei components (Alves et al. 2019). However, this method has never been applied to explore the subcortical anatomy of the VAN and the DAN.*”

3) Introduction. Please motivate diffusion connectivity analysis in the introduction as well, as this seems to be one of the methods used in the study of subcortical networks. Ditto for the neurochemical/transmitter resources.

R: We had introduced diffusion connectivity analysis and neurotransmitter systems analysis in the methods, but we agree that it is more appropriate to do it in the introduction.

The following changes were made:

- Introduction

Added: *“Delineating the subcortical components of the DAN and the VAN would allow us to revisit their underlying circuitry through diffusion-weighted imaging tractography that enables in vivo reconstruction of associative, commissural, and projection white-matter tracts (Behrens et al. 2003; Catani and Thiebaut de Schotten 2012; Zhang et al. 2022). A clearer characterization of the DAN and VAN circuitry will help to better understand brain interactions in healthy and pathological brains (Suárez et al. 2020; Thiebaut de Schotten et al. 2020)”*

Amended and moved from the Methods to the Introduction: *“Subcortical structures also play a critical role within the neurotransmitter systems. Brainstem nuclei are the primary sources of neurotransmitter synthesis and send axonal projections to the cortex and the basal ganglia. The basal ganglia are central targets of the neurotransmitter axonal projections and mediate their physiological effects. Yet the neurochemistry of the DAN and the VAN is limited to primate studies. These studies reported a noradrenergic innervation of regions of the primate attention networks, including the temporo-parietal junction and the frontal lobe (Morrison and Foote 1986; Foote and Morrison 1987; Bouret and Sara 2005). Noradrenaline has been proposed as a critical trigger for the reorientation of attention (Bouret and Sara 2005; Corbetta et al. 2008). However, despite its essential neuroscientific and medical importance (Sanefuji et al. 2017) the neurochemical signatures of the VAN and the DAN have never been contrasted in humans. Such an endeavor is now*

possible thanks to the macroscale mapping of the neurotransmitter receptors and transporters in humans by means of positron emission tomography (PET) and single-photon emission computerized tomography (SPECT) scans (Hansen et al. 2021). Accordingly, a normative atlas of nine neurotransmitter systems aligned in the MNI space is now openly available and allows for the first time for the investigation of the neurochemical signature of brain circuits.”

4) Methods: I am not fully clear on how the functional alignment was done in the current work. The authors mention that VAN/DAN maps were aligned with ANTS based on cross-correlation similarity, but I fail to understand the specifics here and which functional features were specifically used for alignment (connectivity patterns, time series, etc). Further details, and potentially a schematic supplementary figure, would greatly help.

R: To clarify the scripts that were used, the link to ANTs scripts ‘buildtemplateparallel.sh’ and ‘WarpImageMultiTransform’ was provided, and our code applying these scripts is freely available at https://github.com/Pedro-N-Alves/VAN_DAN_functional_alignment.

The functional alignment was made between the 110 individual Pearson’s correlation maps of each network. The performed steps, in more detail, are:

a) The cortical parcellations of VAN and DAN were obtained from the Gordon’s et al. parcellation – 23 VAN parcels and 32 DAN parcels;

b) Each parcel was used as a seed to calculate the Pearson’s correlation maps from the time-series resting-state fMRI in MNI152 space. With this, we obtained 23 VAN and 32 DAN Pearson’s correlation maps for each participant;

c) To obtain the individual VAN and DAN maps, we calculated the median of the 23 VAN and 32 DAN Pearson's correlation maps. With this, we got 110 VAN and 110 DAN individual maps.

d) The 110 VAN maps were functionally aligned with diffeomorphic transformations, using the ANTs scripts 'buildtemplateparallel.sh' and 'WarpImageMultiTransform'.

As suggested, a supplementary figure was created to explain these steps better.

In addition, this section of the methods was edited to detail this information.

The following changes were made:

- Supplementary figure:

Added: “

Supplementary Figure 1. Schematic representation of the steps performed during functional alignment. First, seed-based resting-state functional correlation maps were computed (Pearson's correlation). Second, the median of the Pearson's correlation maps was calculated to obtain the subjects' functional correlation map of the studied network (VAN or DAN). Third, the subjects' functional correlation maps were functionally aligned using diffeomorphic transformations."

- Methods, VAN and DAN maps in the functional space

Amended: *"The 110 individual VAN Pearson's correlation maps in the MNI152 space were aligned in a functional space to optimize their inter-individual alignment of functional areas (Mueller et al. 2013; Robinson et al. 2014; Langs et al. 2015; Glasser et al. 2016)"*

Amended: *"The same steps were performed with the 110 DAN Pearson's correlations maps. A schematic representation of the functional alignment steps is available in Supplementary Figure 1."*

- Code availability statement:

Added: *"The ANTs scripts 'buildtemplateparallel.sh' and 'WarpImageMultiTransform' at <https://github.com/ANTsX/ANTs>; the code where these scripts were applied for the functional alignment performed in this work at https://github.com/Pedro-N-Alves/VAN_DAN_functional_alignment."*

5) The lateralization analysis was interesting, but came a bit out of the blue. I would add further rationale in the introduction as well on why one should

inspect inter-hemispheric asymmetries in the context of these networks and their subcortical components.

R: We added information about the networks' lateralization in the introduction and developed the topic of network asymmetries in the discussion.

The following changes were made:

- Introduction

Amended: *“Classical core regions of the DAN are the intraparietal sulcus, the superior parietal lobe, and the frontal eye fields. The DAN is considered to have no hemispheric lateralization (Corbetta et al. 2000, 2008; Buschman and Miller 2007; Thiebaut de Schotten et al. 2011; Amemiya et al. 2021).”*

Amended: *“In contrast, the temporo-parietal junction and the ventrolateral prefrontal cortex constitute the central regions of the VAN. Evidence demonstrates that the network is right lateralized (Downar et al. 2000; Corbetta et al. 2008; Thiebaut de Schotten et al. 2011).”*

- Discussion

Added (after *“In our analysis, VAN and DAN structural connectivity maps were right-lateralized.”*): *“The right lateralization of the VAN is established in the literature. Evidence demonstrates that the SLF III has a larger volume in the right hemisphere and that its anatomical lateralization correlates with visuomotor processing abilities and the asymmetries of visuospatial task (Thiebaut de Schotten et al. 2011a; Chechlacz et al. 2015; Budisavljevic et al. 2017; Cazzoli and Chechlacz 2017; Howells et al. 2018; Amemiya et al. 2021). The SLF I, the main tract connecting DAN cortical regions, does not show a preferential lateralization (Thiebaut de Schotten et al. 2011; Amemiya et al. 2021). However, some DAN areas*

might be right-lateralized (Bartolomeo and Seidel Malkinson 2019). The right intraparietal sulcus (Sheremata and Silver 2015) and frontal eye field (Szczepanski et al. 2010) increase their activity for both visual fields, while the left preferentially reacts to contralateral stimulations. The processing of both visual fields in the right hemisphere is corroborated by right hemisphere stroke patients with hemispatial neglect who also present with deficits in goal-driven selective attention for ipsilateral stimuli (Snow and Mattingley 2006). Hence, while the cortical extent of the DAN was not asymmetrical, our structural connectivity analysis, including the cortico-subcortical projection tracts, might have the function-specific dimension of the right-lateralization of the DAN.”

6) Results: Figure 1 could benefit from medial cortical views as well.

R: We thank the reviewer for this suggestion and amended figure 1a accordingly.

7) Results. The neurotransmitter associations are interesting, but it needs to be highlighted that these maps come from entirely different datasets with likely

different socio-demographic criteria as HCP. This should be highlighted in abstract and other parts for the manuscripts, and limitations/indirectness of the approach should be acknowledged.

R: The distribution of neurotransmitters' receptors and transporters was obtained from Positron Emission Tomography (PET), and Single-Photon Emission Computerized Tomography (SPECT) scans. It is impossible to map several neurotransmitter systems in the same participant because PET and SPECT tracers are radioactive. Hansen and colleagues collected their data from different samples (of healthy subjects) and multiple research groups. This information was added to the abstract and clarified in the introduction.

Although Hansen and colleagues replicated their findings in an independent autoradiography dataset, we agree that the different provenience of PET and SPECT data is a limitation.

The following changes were made:

- Abstract

Amended: *“These nuclei are densely connected structural network hubs as revealed by diffusion-weighted imaging tractography. Their projections establish interrelations with the acetylcholine nicotinic receptor as well as dopamine and serotonin transporters, as demonstrated in a spatial correlation analysis with a normative atlas of neurotransmitter systems.”*

- Introduction

Amended: *“By collecting more than 1200 positron emission tomography (PET) and single-photon emission computerized tomography (SPECT) scans from different samples of healthy individuals and multiple research groups, they provided a*

normative atlas of nine neurotransmitter systems aligned in the MNI space (Hansen et al. 2021).”

- Discussion

Added: “In addition, the neurotransmitter systems' normative atlas is derived from different samples (Hansen et al. 2021). As PET and SPECT tracers are radioactive, it is not possible to map several neurotransmitter systems in the same participants. Although the atlas was replicated in an independent autoradiography dataset and all scans were acquired in healthy volunteers (Hansen et al. 2021), the heterogeneity of the data sources may represent a limitation for its interpretation.”

8) Please clarify how spatial similarity between neurotransmitter and imaging maps was tested. Did the authors use null models that control for autocorrelation to establish significances?

R: We agree with the importance of generating null models to control for autocorrelation in the establishment of significances in pairwise comparisons of brain maps (Alexander-Bloch et al 2018, Burt et al 2020, Markello and Misic 2021).

However, in this work we inferred statistical significances based on group level analysis, not in pairwise map comparisons. The steps performed were:

a) We used the ‘compare_images’ script (<https://github.com/netneurolab/neuromaps>) to compute the Spearman’s correlation between individual structural connectivity maps and each of the neurotransmitter systems’ density maps. With this, we obtained 110 Spearman’s correlation values for each neurotransmitter system and network.

b) Then, we tested if the distribution of the 110 Spearman's correlation values was significantly higher than zero, by performing a non-parametric statistical analysis (one-sided Wilcoxon test).

c) Finally, the obtained p-values were corrected for multiple comparisons using the Bonferroni correction. As 19 comparisons were performed, the result was considered statistically significant if $p\text{-value} < 0.003$.

We opted for a group level statistical analysis based on individual VAN and DAN structural connectivity distributions, instead of a pairwise statistical comparison between the group average map of VAN and DAN structural connectivity distributions and the neurotransmitter systems' maps, because with this approach we can clearly assess the intersubject variability (plotted in figure 6a). In addition, although null models reduce the probability of spurious findings, they may still inflate false positives and have variable performances in pairwise comparisons (Markello and Misic 2021).

To complement this section, according to the reviewer's comment, we performed a supplemental analysis of pairwise comparisons using null models. The obtained results were similar. They were added to the supplementary materials.

The following changes were made:

- Methods, Structural correlations with the neurotransmitter system

Added: *“We obtained the correlation values distribution between the 110 individual VAN and DAN maps and each neurotransmitter map. To analyze if the obtained distributions (each composed by 110 correlation values) were significantly higher than zero, a non-parametric statistical test was performed (one-sided*

Wilcoxon test). The obtained p-values were corrected for multiple comparisons using the Bonferroni correction.”

Amended: “Finally, we analyzed whether the correlation distributions were different between VAN and DAN, as well as if they were different between hemispheres (paired t-test or Wilcoxon test, according to data distribution; the Bonferroni correction was also applied).”

Added: “A supplemental pairwise analysis was performed. The average map of the 110 individual VAN and DAN structural projection maps was correlated with the neurotransmitter maps (Spearman’s correlation; neuromaps’ tool ‘compare_images’; Markello et al. 2022; <https://netneurolab.github.io/neuromaps/>). To control for spatial autocorrelations and reduce the risk of false positive results, statistical significance was inferred based on null models generation (Alexander-Bloch et al. 2018; Burt et al. 2020; Markello and Misic 2021). Volumetric data was parcellated according to the Automated Anatomical Labeling atlas 3 (AAL3; Rolls et al. 2020), using the neuromaps’ utility ‘Parcellater’ (Markello et al. 2022; <https://netneurolab.github.io/neuromaps/>). AAL3 was chosen because it includes cortical and subcortical parcels. The null parcellations were generated from the average VAN and DAN structural projection maps using the neuromaps’ function ‘nulls.burt2020’ (5000 permutations, generating 5000 null parcellations; Burt et al. 2020; Markello et al. 2022; <https://netneurolab.github.io/neuromaps/>).”

- Results, Correlation with the neurotransmitter system

Added: “The supplemental pairwise correlation analyses between the average VAN and DAN structural projection maps and the neurotransmitter maps revealed similar results: VAN had a significant positive spatial correlation with acetylcholine

$\alpha 4\beta 2$ nicotinic receptors and acetylcholine, dopamine, noradrenaline and serotonin transporters (Supplementary Table 5); DAN had a significant positive spatial correlation with acetylcholine $\alpha 4\beta 2$ nicotinic receptors and acetylcholine, dopamine and noradrenaline transporters (Supplementary Table 6)."

- Figure 6 legend

Amended: "*Correlation between the structural projections of the brainstem nuclei and the neurotransmitter systems. (a) Distributions of the Spearman's correlations for the available maps of neurotransmitter receptors and transporters*"

- Supplementary material

Added: "*Table 5. Pairwise correlations between neurotransmitter maps and the average VAN structural projection map.*"

Added: "*Table 6. Pairwise correlations between neurotransmitter maps and the average DAN structural projection map.*"

We hope we have addressed all the queries appropriately.

We thank the reviewers for their comments and remain fully available for further clarifications.

REVIEWERS' COMMENTS:

Reviewer #2 (Remarks to the Author):

Having access to scripts is beneficial for the scientific community, and I am glad to see this added. There are now sufficient details in the graph analysis description to allow it to be replicated. The discussion was improved by including a discussion of interpreting graph theory analysis results. A thorough explanation of VAN and DAN's lateralization is also provided by the authors.

The authors have addressed all my concerns adequately.

Reviewer #3 (Remarks to the Author):

I thank the authors for incorporating my suggestions, and recommend the paper for publication.